# Association of Possible Sarcopenia or Sarcopenia with Body Composition, Nutritional Intakes, Serum Vitamin D Levels, and Physical Activity among Patients with Type 2 Diabetes Mellitus in Taiwan

**DOI:** 10.3390/nu15183892

**Published:** 2023-09-07

**Authors:** Yu-Ting Hsu, Jian-Yu Lin, Chien-Ju Lin, Yau-Jiunn Lee, Wen-Hsin Chang

**Affiliations:** 1Nutrition and Food Service Department, Kaohsiung Veterans General Hospital, No. 386, Dazhong 1st Rd., Zuoying Dist., Kaohsiung City 81362, Taiwan; vickyhsu0428@gmail.com (Y.-T.H.); linchainyu0822@gmail.com (J.-Y.L.); 2Department of Sports Medicine, College of Medicine, Kaohsiung Medical University, No. 100, Shih-Chuan 1st Road, Sanmin Dist., Kaohsiung City 80708, Taiwan; 3School of Pharmacy, College of Pharmacy, Kaohsiung Medical University, No. 100, Shih-Chuan 1st Road, Sanmin Dist., Kaohsiung City 80708, Taiwan; mistylin@kmu.edu.tw; 4Lee’s Endocrinology Clinic, No. 396, Guangdong Road, Pingtung City 90028, Taiwan; lee@leesclinic.org

**Keywords:** type 2 diabetes mellitus, sarcopenia, serum vitamin D levels, nutritional intake, physical performance

## Abstract

This study estimates the association between sarcopenia and blood biochemical parameters, nutritional intake, anthropometric measurements, physical performance, and physical activity in patients with type 2 diabetes mellitus (T2DM). Participants were recruited from a primary care clinic in Kaohsiung City. According to the diagnosis criteria of the Asian Working Group for Sarcopenia (AWGS) in 2019, 110 patients with T2DM (aged 50–80 years) were divided into three groups: non-sarcopenia (*n* = 38), possible sarcopenia (*n* = 31), and sarcopenia (*n* = 41). Blood samples were collected, and nutritional intake was evaluated by a registered dietitian. A food frequency questionnaire and a Godin leisure-time exercise questionnaire were used to assess their daily vitamin D intake and physical activity. There were significant differences in age, serum vitamin D levels, nutritional intake, anthropometric measurements, and physical performance between the three groups. In elderly patients with T2DM, reduced serum 25-hydroxyvitamin D [25(OH)D] levels and daily energy intake were significantly associated with possible sarcopenia. Age, lower BMI, reduced serum 25(OH)D, and reduced dietary protein and vitamin D intake were significantly associated with sarcopenia. These findings may serve as the basis for intervention trials to reduce the prevalence of sarcopenia.

## 1. Introduction

Sarcopenia is a geriatric syndrome characterized by progressive and generalized loss of skeletal muscle mass, strength, and function, and it is correlated with physical disability, poor quality of life, and death [1]. Sarcopenia was also officially recognized as a disease in 2016 (International Diagnostic Code of Diseases M62.84). According to the Health ABC Generation Study, elderly people with diabetes show decreased muscle mass and strength in the lower extremities compared to healthy people of the same age [2]. Therefore, the Asian Working Group for Sarcopenia (AWGS) has specifically suggested that patients with type 2 diabetes mellitus (T2DM) be screened for sarcopenia [1].

Sarcopenia in patients with T2DM is associated with reduced quality of life, a lower physical activity level, and poor nutrition [3]. A previous study has also found that a lower serum vitamin D level is associated with lower muscle mass, lower muscle strength, poorer physical performance, and a higher risk of sarcopenia [4]. Elderly people may have a decreased serum vitamin D level because of factors such as reduced dietary intake, insufficient sun exposure, thinner skin, and decreased intestinal absorption. Therefore, elderly people with diabetes and vitamin D deficiency may be more susceptible to sarcopenia.

Despite growing interest in diabetic sarcopenia in Taiwan, a limited number of studies have investigated the risk factors for sarcopenia in patients with T2DM. Moreover, most of these epidemiological studies did not observe the effects of nutritional intake, serum vitamin D level, or physical activity on the risk of sarcopenia. Furthermore, unlike the diagnostic criteria used by other studies, the present study used the latest 2019 AWGS guidelines [5] as diagnostic criteria for sarcopenia. This study aimed (i) to compare the differences in blood biochemical values, nutritional intake, physical performance, and physical activity between patients with and without sarcopenia and (ii) to determine the risk factors of possible sarcopenia or sarcopenia among patients with T2DM in Taiwan.

## 2. Materials and Methods

### 2.1. Participants

The setting was a primary care clinic-based study in southern Taiwan. We recruited participants aged 50–80 years from August 2020 to October 2020 who had received the diagnosis of T2DM more than 1 year before the study’s initiation. Patients with a musculoskeletal injury (such as a sprain, contusion, bruise, or fracture) or a major medical condition (such as chronic kidney disease (CKD), diabetic nephropathy, a history of cancer, heart disease, and stroke) which with known risks for sarcopenia, those undergoing dialysis, having a coronary stent (to avoid any electromagnetic interference), being unable to stand or walk alone or with accessories, having a cognitive impairment, and being involved in a weight loss program in the preceding 3 months (such as consuming a calorie-restricted diet, taking weight loss-related medications, or having lost more than 5% of body weight) were excluded.

A convenience sample of 110 subjects was enrolled from a clinic population of diabetics over 50 years of age. Participants consented to complete a general questionnaire and a physical activity questionnaire, which included questions related to sex, age, diabetes onset period, lifestyle habits (smoking, alcohol drinking, and betel nut consumption), medications used in the treatment of chronic diseases (such as oral hypoglycemic agents, insulin injections, blood-pressure-lowering agents, and hypolipidemic agents), use of nutritional supplements and vitamin D supplements, daily sun exposure (frequency and time) and sun protection, and daily leisure-time physical activity. Then, 5 mL of routine venous blood was drawn for biochemical analysis. Body composition and physical performance were measured for the diagnosis of sarcopenia, and each participant’s dietary patterns were assessed by a registered dietitian.

This study was approved by the Human Research Ethics Review Committee of Kaohsiung Veterans General Hospital (IRB: KSVGH20-CT8-12). All participants were informed in detail of the research purpose, process, and precautions before they participated, and they were enrolled only after their consent was obtained. The study was conducted according to the Declaration of Helsinki.

### 2.2. Anthropometric Measurements

Participants were asked to empty their bladders before the anthropometric measurements. The participants’ heights were determined using an automatic measuring machine height weight scale (Super-View HW-3050, Fubio Medical Systems Co., Ltd., Taoyuan, Taiwan); their waist and hip circumferences were evaluated using a measuring tape; and body weight, BF%, and BMI were obtained using the bioelectrical impedance analysis (BIA) method by an Inbody 270 body composition analyzer (Biospace Co., Ltd., Seoul, Republic of Korea). Systolic blood pressure and diastolic blood pressure were measured using an electronic blood pressure monitor (HBP-9020, Omron Healthcare Co., Ltd., Kyoto, Japan).

### 2.3. Screening for Sarcopenia

As suggested in the AWGS 2019 guideline, calf circumference measurement [6] and sarcopenia self-assessment scales (e.g., the SARC-F scale [7] and the SARC-CalF scale [8]) are practical tools for sarcopenia risk assessment. In our primary care clinics, calf circumference measurement and the SARC-F scale were used for sarcopenia risk assessment.

Calf circumference was measured with the participant in a sitting posture. The participant was asked to remove clothing covering the calf and to place both feet on the ground naturally, with the calf and thigh at a 90° angle. The tape measure was kept horizontal as it was passed around the widest part of the calf, and the circumference was measured without squeezing the skin.

The SARC-F scale was completed by the participants; this scale comprises five assessment items: strength, assistance in walking, rising from a chair, climbing stairs, and falls. Each item has a score of 0–2 points, and the total score is 0–10 points. A total score of ≥4 points indicates that the individual is at risk of sarcopenia; higher scores indicate a higher sarcopenia risk.

### 2.4. Diagnosis of Sarcopenia

Following the latest 2019 AWGS diagnostic consensus on sarcopenia [5], BIA (Inbody 270) was used as a quick and reliable tool in community settings [9] for the assessment of skeletal muscle mass index (SMI), which was calculated by appendicular skeletal muscle mass (ASM)/height^2^ and ASM/BMI ratio [5,10,11].

A handgrip strength test with a digital handheld dynamometer (Camry Scale, South El Monte, CA, USA) was used for the assessment of muscle strength. The participant was asked to adopt a standing position with their elbows completely relaxed and straight. Their hands were tested twice each, and the maximum grip strength value was considered to be the muscle strength test value. Low muscle strength was defined as a handgrip strength of <28 kg for males and <18 kg for females [12].

The physical performance test included a short physical performance battery (SPPB), which comprises a 4-m gait speed test, a five-times-sit-to-stand test, and a balance test. The total score was 0 (worst performance) to 12 (best performance) points. A score < 9 was indicative of low physical performance [13]. In the balance test, the participant was asked to stand with their feet side by side, one foot against the side of the big toe of the other foot, and their feet in a straight line. Whether the participant could maintain each position for 10 s was evaluated. In the 4-m gait speed test, the participant was asked to walk in a straight line for 4 m. The time required to complete this walk was recorded, and the walking speed was then calculated. Slow gait speed was defined using the AWGS 2019 reference value of <1.0 m/s. In the five-times-sit-to-stand test, the participant was asked to sit on a chair, stand up, and sit back again five times. The time taken to complete this task was recorded in seconds. A longer time (≥12 s) was indicative of low physical performance [14].

The key diagnosis criteria for “possible sarcopenia” according to AWGS 2019 are low muscle strength (male grip strength < 28 kg; female grip strength < 18 kg) with or without reduced physical performance. “Sarcopenia” is diagnosed when low muscle mass plus low muscle strength or/and low physical performance are detected (male SMI < 7.0 kg/m^2^, grip strength < 28 kg or/and gait speed < 1 m/s or five-times sit-to-stand test ≥ 12 s; female SMI < 5.7 kg/m^2^, grip strength < 18 kg or/and gait speed <1 m/s or five-times sit-to-stand test ≥ 12 s). Other participants without any low muscle mass, low muscle strength, or low physical performance were classified as “non-sarcopenia”.

### 2.5. Biochemical Analyses

Routine blood tests for T2DM—including fasting blood sugar, serum HbA1c, triglyceride (TG), total cholesterol (TC), high-density lipoprotein cholesterol (HDL-C), low-density lipoprotein cholesterol (LDL-C), serum creatinine (S-Cr), and serum 25(OH)D concentration—were analyzed. The blood samples were entrusted to the Zhongshan Medical Laboratory of Pingtung City, Taiwan, which conducted a unified blood biochemical test.

The simplified Modification of Diet in Renal Disease formula (=186 × S-Cr − 1.154 × Age − 0.203 × 0.742 (if female patient)) was used to calculate the estimated glomerular filtration rate (eGFR), which was used as the renal function index.

The Roche Immunoassay Total Vitamin D Assay Reagent (Elecsys Vitamin D total, Roche Diagnostics GmbH, Mannheim, Germany) and electrochemiluminescence immunoassay were used to quantify the serum 25(OH)D concentration, reflecting total body vitamin D stores.

### 2.6. Nutritional Intake

Nutritional intake was assessed using a 24-h dietary recall and food frequency questionnaire (FFQ). A professional dietitian conducted a one-to-one inquiry for the 24-h dietary recall, and the FFQ was administered to collect information regarding the intake of vitamin-D-containing foods in the preceding month.

#### 2.6.1. 24-h Dietary Recall

Dietary intake data were converted into total daily calorie intake (kcal), and macronutrients (proteins, fats, and carbohydrates) were expressed in grams (g) and grams per kilogram (g/kg) of body weight (BW).

#### 2.6.2. Food Frequency Questionnaire

The food sources of FFQ were developed by the Nutrients Database of the United States Department of Agriculture (USDA) and the Food and Drug Administration, Ministry of Health and Welfare in Taiwan. This questionnaire contains questions about the frequency and portion size of consumption of particular items to calculate the daily intake of vitamin D. In this study, Cronbach’s α was used to measure the internal consistency of the items included in the questionnaire. The result of the reliability measure was acceptable: α = 0.715 in our study, which means all items in our questionnaire were internally consistent and reliable to assess the daily intake of vitamin D.

### 2.7. Leisure-Time Physical Activity Measurement

The Godin leisure-time exercise questionnaire, developed by Godin and Shephard in 1985 [15], was used to assess the “daily leisure-time physical activity” of the participants. The 4-item questionnaire, with the first three questions seeking information on the number of times one engages in different intensities (high, moderate, and light) of leisure-time physical activity for at least 15 min for 7 days. The number of times a participant engaged in physical activities of different intensities was included in the formula to calculate the total score (total score = [high-intensity activity × 9] + [moderate-intensity activity × 5] + light activity × 3]).

### 2.8. Statistical Analysis

Statistical analysis was performed with the SPSS Version 22.0 statistic software package (IBM SPSS Inc., Chicago, IL, USA). Categorical variables are expressed as frequencies and percentages, and continuous variables are expressed as mean and standard deviation (SD) when data follows a normal distribution or as median and interquartile range (IQR = 25–75th percentile) when the distribution departs from normality. In each group, normality was tested using the Kolmogorov–Smirnov test. The percentages were compared, as appropriate, using the Chi-square test or the exact Fisher test.

Given the low sample size, the sarcopenia and possible sarcopenia groups were merged into a single category. Then, a logistic model for the outcome of possible sarcopenia or sarcopenia versus non-sarcopenia was estimated. A univariate analysis was performed to explore the factors associated with possible sarcopenia or sarcopenia. Significant variables were entered into the multivariate logistic regression model with the stepwise method and checked for multicollinearity and a Hosmer–Lemeshow test was performed to determine the factors associated with possible sarcopenia or sarcopenia in the patients with T2DM. The Nagelkerke R square was used to evaluate the predictive accuracy of the logistic regression model. Results were expressed as an odds ratio (OR) with a corresponding 95% confidence interval (CI). Statistical significance was set at *p* < 0.05.

The sample size was calculated using G*Power version 3.1.9.7, with the F test set α error at 0.05, power level at 90%, and a theoretical large effect size of 0.4. Since the main objective was to compare the differences in parameters between the three groups, a one-way ANOVA was performed to calculate the sample size. The results showed that the minimum number of participants for this study was 84. With additional compensation for a possible dropout rate of 30%, the sample size was set at 110 participants.

## 3. Results

The study enrolled 110 participants (46 males and 64 females), including 38 without sarcopenia, 31 with possible sarcopenia (28.2%), and 41 with sarcopenia (37.3%) (Figure 1); their mean ages were 64.4 ± 7.3, 67.1 ± 7.2, and 70.2 ± 6.0 years, respectively. The mean age of the participants in the sarcopenia group (70.2 ± 6.0 years) was significantly higher than that in the non-sarcopenia group (64.4 ± 7.3 years), and differences were noted in gender, diabetes onset period, and smoking habits. No significant differences were discovered in the use of chronic disease medications, alcohol and betel nut consumption, or chronic diseases (diabetes, hypertension, and hyperlipidemia; Table 1).

The main sources of vitamin D in humans are diet and sunlight exposure; therefore, this study also assessed daily sun exposure and sun protection. A comparison of the daily sun exposure and sun protection status showed no significant difference between the groups (Table 2). However, the results found that 67.3% of the patients had daily sun exposure for more than 20 min (26 (68.4%) in the non-sarcopenia group, 22 (71.0%) in the possible sarcopenia group, and 26 (63.4%) in the sarcopenia group).

There were significantly higher cholesterol levels in the sarcopenia group than in the possible sarcopenia group and higher HDL-C levels than in the non-sarcopenia group. The sarcopenia and possible sarcopenia groups had significantly lower serum 25(OH)D levels than the non-sarcopenia group (Table 3).

The total consumption of calories and macronutrients (carbohydrates, proteins, and fats) differed significantly among the three groups. The calorie intake per kilogram of BW was significantly lower in the possible sarcopenia group (25.1 ± 4.2 kcal/kg BW) than in the non-sarcopenia group (27.9 ± 4.6 kcal/kg BW). The protein intake per kilogram of BW was significantly lower in the sarcopenia group (0.89 ± 0.19 g/kg) than in the possible sarcopenia (0.92 ± 0.20 g/kg BW) and non-sarcopenia groups (1.04 ± 0.25 g/kg BW), and carbohydrate and lipid intake were also significantly lower in the sarcopenia group. In addition, the FFQ revealed that the sarcopenia group had the lowest daily intake of vitamin-D-rich foods than the other two groups, but no significant difference between the groups was discovered (Table 4).

Regarding physical performance, the body composition (body weight, BMI, BF%, and waist and hip circumference), diagnostic indicators of sarcopenia (muscle mass, muscle strength, and physical performance), and sarcopenia risk assessment (calf circumference and SARC-F scale) were significantly worse in the Sarcopenia group than the other two groups. In addition, although no significant difference in daily leisure-time physical activity was discovered among the three groups, the score of leisure-time physical activity in the possible sarcopenia and sarcopenia groups was lower than that of the non-sarcopenia group (*p* = 0.062) (Table 5).

Univariate logistic regression analyses showed that diastolic blood pressure; daily energy, protein, and fat intake; and serum 25(OH)D levels were associated with possible sarcopenia. Age, body weight; BMI; waist and hip circumference; serum TC levels; serum HDL-C levels; serum 25(OH)D levels; daily energy, carbohydrate, protein, fat, and vitamin D intake; and physical activity scores were also associated with sarcopenia. Table 6 shows the selected risk factors associated with possible sarcopenia or sarcopenia using the multivariate logistic regression model with the stepwise method. Compared with the non-sarcopenia group, dietary energy intake (OR = 0.877; 95% CI, 0.774–0.994; *p* = 0.040) and serum 25(OH)D levels (OR = 0.927; 95% CI, 0.863–0.997; *p* = 0.040) were risk factors for possible sarcopenia. Age (OR = 1.231; 95% CI, 1.071–1.415; *p* = 0.004), lower BMI (OR = 0.438, 95% CI: 0.283–0.680; *p* = 0.000), serum 25(OH)D levels (OR = 0.901, 95% CI: 0.812–0.999; *p* = 0.048), dietary protein (OR = 0.001, 95% CI: 0.000–0.166; *p* = 0.009), and vitamin D intake (OR = 0.915, 95% CI: 0.849–0.986; *p* = 0.020) were risk factors for sarcopenia.

## 4. Discussion

This is one of the few studies in Taiwan to use the 2019 AWGS diagnostic criteria to define sarcopenia and to compare the nutritional intake (including assessment of dietary vitamin D intake and sun exposure), blood biochemical parameters (especially serum vitamin D concentration), body composition, leisure-time physical activity, and other related factors in elderly patients with T2DM and possible sarcopenia or sarcopenia. The main finding of this study was that the risk factors for possible sarcopenia were daily energy intake and serum 25(OH)D levels, while the risk factors for sarcopenia were age, BMI, dietary protein and vitamin D intake, and serum 25(OH)D levels among patients with T2DM.

T2DM was independently associated with an increased risk of sarcopenia, suggesting that patients with T2DM should be considered in screening for sarcopenia [16]. HbA1c can reflect the long-term glycemic control from the last 3 months; the present study found no significant difference in HbA1c level among the three groups with T2DM, which is inconsistent with previous studies [17,18]. The difference between those studies and the present study may be due to our study design; only a single HbA1c data point was collected, and the participants regularly used hypoglycemic drugs to control their blood glucose levels. Therefore, the results may not truly reflect their glycemic status. According to the United Kingdom Prospective Diabetes Study and other epidemiological studies, people with dyslipidemia are more prone to cardiovascular diseases than those without dyslipidemia, regardless of whether they have diabetes [19]. Clinically, patients with T2DM should pay attention to whether their lipid panel blood tests are abnormal and use lipid-lowering drugs to prevent cardiovascular diseases. In this study, although the TC concentration was significantly higher in the sarcopenia group than in the possible sarcopenia group, most patients were routinely taking lipid-lowering drugs; therefore, the values of the lipid panel were within the normal range.

Chuang et al. found that regardless of age and gender, serum vitamin D concentration was strongly correlated with muscle mass and physical performance in Taiwan [20]. However, the mean serum 25(OH)D concentrations in the three groups indicated that our participants achieved vitamin D sufficiency levels (defined as ≥30 ng/mL) [21], and the findings were consistent with a previous study showing that vitamin D sufficiency may be associated with prolonged sun exposure in elderly men also living in southern Taiwan [22]. The research period of this study was August to October, and 67.3% of the participants reported long exposure (≥20 min) to the sun. We also found that lower dietary vitamin D intake was a risk factor for sarcopenia. The dietary reference intake (DRI) in Taiwan determines the adequate intake (AI) for vitamin D, which was recently reviewed upwards from 10 µg/day to 15 µg/day for those over 50 years. However, this value is still lower than in other countries. According to the International Osteoporosis Foundation, dietary vitamin D intake of 20 to 25 µg/day is required to prevent both falls and bone fractures in the elderly [23]. Therefore, future research may be needed to determine the optimal recommended intake of vitamin D among elders with T2DM and sarcopenia.

Our study indicated that higher BMI had a protective effect on sarcopenia, which is consistent with previous studies in elderly Koreans or elderly Japanese diabetes patients [23,24,25]. Furthermore, a previous study indicated that obesity was a protective factor for sarcopenia when defined by BF% instead of BMI [26]. Sarcopenic obesity (SO) is a category of obesity and a high-risk geriatric syndrome in the elderly. Sedentary behavior and an unhealthy diet are independently associated with SO [27,28]. Although there was no significant difference in daily leisure-time physical activity between the three groups in the present study, T2DM patients with possible sarcopenia or sarcopenia were less physically active than those without sarcopenia, which is consistent with the previous study [3]. The present study also found that lower protein intake (g/kg BW) is a risk factor for T2DM complicated with sarcopenia, which is consistent with previous studies [29,30]. More large-scale local studies are required to further explore the recommended daily protein intake for preventing sarcopenia in this population.

The strengths of this study are that it is one of the very few studies in Taiwan to examine the association between possible sarcopenia or sarcopenia and multiple clinical factors among elderly patients with T2DM. We investigated as comprehensively as possible the association between blood biochemical parameters (especially serum 25(OH)D concentration), nutritional intake (especially dietary vitamin D intake and supplements), physical function, and physical activity (Godin leisure-time exercise questionnaire) in these patients. In addition, a professional dietitian used the 24-h dietary recall and FFQ to assess the actual daily intake of vitamin D.

This study also has some limitations. First, we could not fully clarify the causal relationship between sarcopenia and its associated factors due to the cross-sectional study design. Second, because we set strict inclusion criteria, the sample size was relatively small. T2DM in the elderly had a higher prevalence of many comorbidities, such as CKD, which is one of the major causes and exacerbating factors for sarcopenia [31]. Third, our primary care clinic has limited space and funding, so the dual-energy X-ray absorptiometry (DXA) and 6-m gait speed tests could not be conducted. Fourth, we cannot rule out the potential recall bias because reliability and accuracy cannot be objectively ascertained in the self-reported variables. Fifth, we only recorded the use of oral medications and insulin treatment in participants; the correlation between specific drugs and sarcopenia was not fully explored. Finally, the data from a local study site (from one city in Taiwan) would not reflect the nationwide prevalence of sarcopenia.

## 5. Conclusions

Sarcopenia is an emerging health problem in the aging diabetic population in Taiwan. We found that possible sarcopenia was associated with reduced serum 25(OH)D levels and lower energy intake. Aging and lower BMI, dietary protein and vitamin D intake, and serum 25(OH)D levels were associated with sarcopenia. These findings may serve as a basis for intervention trials to reduce the prevalence of sarcopenia.

## Figures and Tables

**Figure 1 nutrients-15-03892-f001:**
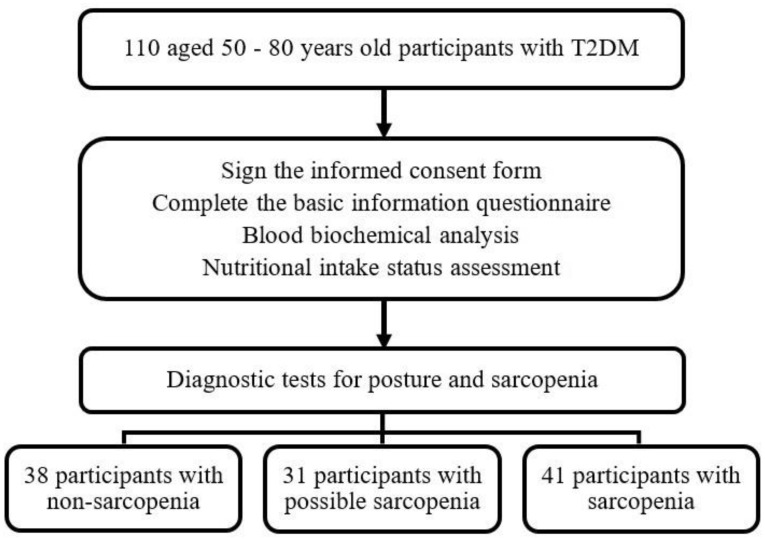
Enrollment diagram. Number of individuals analyzed for the main outcome of non-sarcopenia, possible sarcopenia, and sarcopenia in participants with T2DM.

**Table 1 nutrients-15-03892-t001:** Comparison of baseline characteristics of the non-sarcopenia, possible sarcopenia, and sarcopenia groups of participants with T2DM ^1,2^.

Variables	Non-Sarcopenia (*n* = 38)	Possible Sarcopenia (*n* = 31)	Sarcopenia (*n* = 41)	*p*
Gender				0.115
Female	17 (44.7%)	20 (64.5%)	27 (65.9%)	
Male	21 (55.3%)	11 (35.5%)	14 (34.1%)	
Age (year)	64.4 ± 7.3 ^a^	67.1 ± 7.2 ^a,b^	70.2 ± 6.0 ^b^	0.001
Duration of Diabetes (year)	12.0 (7.0, 15.3)	14.0 (7.0, 16.0)	14.0 (11.0, 19.0)	0.122
Smoking	10 (26.3%)	3 (9.7%)	8 (19.5%)	0.212 *
Alcohol drinking	9 (23.7%)	3 (9.7%)	6 (14.6%)	0.300 *
Betel nut	5 (13.2%)	2 (6.5%)	1 (2.4%)	0.185 *
Oral hypoglycemic medications	37 (97.4%)	29 (93.6%)	39 (95.1%)	0.744
Insulin injection	7 (18.4%)	1 (3.2%)	9 (22.0%)	0.064 *
Antihypertensive agents	17 (44.7%)	11 (35.5%)	14 (34.2%)	0.586
Hypolipidemic agents	33 (86.8%)	27 (87.1%)	33 (80.5%)	0.662

^1^ Values are mean ± SD, median (IOR), or frequencies (%). Different superscript letters (^a^, ^b^) indicate significant differences compared with the other groups, as assessed using one-way ANOVA (normal distribution) and the Kruskal–Wallis test (non-normal distribution) (*p* < 0.05). The chi-square test, or Fisher’s exact test (*), was used for categorical variables. ^2^ T2DM: type 2 diabetes mellitus.

**Table 2 nutrients-15-03892-t002:** Comparison of frequency and time of sun exposure and sun protection in the non-sarcopenia, possible sarcopenia, and sarcopenia groups of participants with T2DM ^1,2^.

Variables	Non-Sarcopenia (*n* = 38)	Possible Sarcopenia (*n* = 31)	Sarcopenia (*n* = 41)	*p*
Sun exposure (day/week)	5.4 ± 2.3	5.0 ± 2.7	4.9 ± 2.7	0.709
Sun exposure (time/day)				0.857 *
No	3 (7.9%)	4 (12.9%)	7 (17.1%)	
<20 min	9 (23.7%)	5 (16.1%)	8 (19.5%)	
20 min–1 h	18 (47.4%)	14 (45.2%)	15 (36.6%)	
>1 h	8 (21.1%)	8 (25.8%)	11 (26.8%)	
Sun protection				0.657
No	15 (39.5%)	10 (24.4%)	17 (41.5%)	
1	10 (26.3%)	12 (38.7%)	15 (36.6%)	
≥2	13 (34.2%)	9 (29.0%)	9 (22.0%)	

^1^ Values are mean ± SD or frequencies (%). One-way ANOVA with Scheffé post hoc test was used for continuous variables. The chi-square test, or Fisher’s exact test (*), was used for categorical variables. ^2^ T2DM: type 2 diabetes mellitus.

**Table 3 nutrients-15-03892-t003:** Comparison of blood biochemical parameters of the non-sarcopenia, possible sarcopenia, and sarcopenia groups of participants with T2DM ^1,2^.

Variables	Non-Sarcopenia (*n* = 38)	Possible Sarcopenia (*n* = 31)	Sarcopenia (*n* = 41)	*p*
FBS (mg/dL)	128.5 (120.0, 143.8)	134.0 (123.0, 145.0)	139.0 (117.5, 154.5)	0.428
HbA1c (%)	7.1 (6.7, 7.6)	7.2 (6.7, 7.7)	7.2 (6.7, 7.8)	0.762
TG (mg/dL)	95.5 (71.8, 162.5)	96.0 (72.0, 127.0)	86.0 (59.5, 122.0)	0.293
TC (mg/dL)	149.7 ± 30.9 ^a,b^	146.9 ± 33.2 ^a^	166.2 ± 31.6 ^b^	0.019
HDL-C (mg/dL)	58.5 (47.8, 63.5) ^a^	59.0 (53.0, 74.0) ^a,b^	64.0 (58.0, 74.0) ^b^	0.019
LDL-C (mg/dL)	76.3 ± 24.3	72.8 ± 18.0	86.2 ± 27.3	0.048
eGFR (mL/min/1.73 m^2^)	78.8 (69.9, 87.2)	75.8 (69.7, 85.4)	75.9 (69.5, 89.5)	0.822
Serum 25(OH)D (ng/mL)	38.4 ± 9.3 ^a^	33.3 ± 5.8 ^b^	33.2 ± 6.7 ^b^	0.004

^1^ Values are mean ± SD, median (IOR), or frequencies (%). Different superscript letters (^a^, ^b^) indicate significant differences compared with the other groups, as assessed using one-way ANOVA (normal distribution) and the Kruskal–Wallis test (non-normal distribution) (*p* < 0.05). ^2^ T2DM: type 2 diabetes mellitus; FBS: fasting blood sugar; HbA1c: glycated hemoglobin; TG: triglyceride; TC: total cholesterol; HDL-C: high-density lipoprotein cholesterol; LDL-C: low-density lipoprotein cholesterol; eGFR: estimated glomerular filtration rate; 25(OH)D: 25-hydroxyvitamin D.

**Table 4 nutrients-15-03892-t004:** Comparison of nutritional intake and supplement use of the non-sarcopenia, possible sarcopenia, and sarcopenia groups of participants with T2DM ^1,2^.

Parameters	Non-Sarcopenia (*n* = 38)	Possible Sarcopenia (*n* = 31)	Sarcopenia (*n* = 41)	*p*
Energy (kcal)	1820.8 ± 292.4 ^a^	1620.1 ± 217.0 ^b^	1411.3 ± 201.1 ^c^	<0.001
Energy (kcal/kg BW)	27.9 ± 4.6 ^a^	25.1 ± 4.2 ^b^	26.4 ± 3.6 ^a,b^	0.019
Carbohydrate (g)	213.8 ± 37.2 ^a^	202.0 ± 41.8 ^a^	174.9 ± 31.6 ^b^	<0.001
Protein (g)	67.9 ± 13.9 ^a^	59.6 ± 11.9 ^b^	47.2 ± 9.9 ^c^	<0.001
Protein (g/kg BW)	1.04 ± 0.25 ^a^	0.92 ± 0.20 ^a,b^	0.89 ± 0.19 ^b^	0.004
Fat (g)	73.4 (63.0, 88.0) ^a^	64.4 (57.5, 74.0) ^b^	55.0 (49.7, 64.0) ^c^	<0.001
Vitamin D (μg)	23.6 (16.7, 36.4)	21.8 (6.2, 40.6)	16.9 (7.5, 24.8)	0.107
Nutritional supplements				
No	21 (55.3%)	14 (45.2%)	19 (46.3%)	0.821
1	9 (23.7%)	7 (22.6%)	9 (22.0%)	
≥2	8 (21.1%)	10 (32.3%)	13 (31.7%)	
Vitamin D supplements	3 (7.9%)	7 (22.6%)	6 (14.6%)	0.237 *

^1^ Values are mean ± SD, median (IOR), or frequencies (%). Different superscript letters (^a^, ^b^, ^c^) indicate significant differences compared with the other groups, as assessed using one-way ANOVA (normal distribution) and the Kruskal–Wallis test (non-normal distribution) (*p* < 0.05). The chi-square test, or Fisher’s exact test (*), was used for categorical variables. ^2^ T2DM: type 2 diabetes mellitus; BW: body weight.

**Table 5 nutrients-15-03892-t005:** Comparison of anthropometric measurements, physical performance, and physical activity of the non-sarcopenia, possible sarcopenia, and sarcopenia groups of participants with T2DM ^1,2^.

Anthropometrics	Non-Sarcopenia (*n* = 38)	Possible Sarcopenia (*n* = 31)	Sarcopenia (*n* = 41)	*p*
Body weight (kg)	65.8 ± 8.6 ^a^	65.5 ± 8.5 ^a^	53.7 ± 6.1 ^b^	<0.001
BMI (kg/m^2^)	25.4 ± 2.7 ^a^	26.6 ± 3.1 ^a^	22.3 ± 2.7 ^b^	<0.001
Waist circumference (cm)	88.0 ± 7.1 ^a^	89.4 ± 9.3 ^a^	81.6 ± 6.7 ^b^	<0.001
Hip circumference (cm)	95.6 ± 5.0 ^a^	96.8 ± 5.9 ^a^	91.2 ± 5.7 ^b^	<0.001
SBP (mmHg)	131.7 ± 15.5	128.3 ± 14.8	128.5 ± 16.3	0.564
DBP (mmHg)	76.0 (71.0, 86.0)	72.0 (66.0, 78.0)	74.0 (64.5, 82.5)	0.065
**Muscle Mass, Strength and Performance**				
BF%	31.1 ± 7.1 ^a^	35.6 ± 6.5 ^b^	32.3 ± 8.0 ^a,b^	0.041
SMI (ASM/Height^2^) (kg/m^2^)	7.0 ± 0.8 ^a^	6.7 ± 0.8 ^a^	5.6 ± 0.7 ^b^	<0.001
SMI (ASM/BMI) (m^2^)	0.73 ± 0.16 ^a^	0.63 ± 0.12 ^b^	0.62 ± 0.15 ^b^	0.001
HGS (kg)	30.5 (22.5, 36.0) ^a^	21.9 (18.9, 25.7) ^b^	21.2 (17.3, 27.0) ^b^	<0.001
SPPB (score)	11.4 ± 0.8 ^a^	9.8 ± 1.5 ^b^	9.3 ± 2.1 ^b^	<0.001
5-time chair stand test (s)	9.3 (8.2, 10.5) ^a^	12.3 (9.9, 13.6) ^b^	12.7 (11.5, 13.4) ^b^	<0.001
CC (right) (cm)	35.7 ± 2.3 ^a^	34.8 ± 2.5 ^a^	31.6 ± 2.0 ^b^	<0.001
CC (left) (cm)	35.7 ± 2.3 ^a^	34.8 ± 2.7 ^a^	31.6 ± 2.0 ^b^	<0.001
SARC-F (score)	0.13 ± 0.41 ^a^	0.58 ± 1.41 ^a,b^	0.95 ± 1.84 ^b^	0.033
PA (score)	21.0 (9.8, 35.0)	12.0 (0.0, 21.0)	15.0 (0.0, 21.0)	0.062

^1^ Values are mean ± SD or median (IOR). Different superscript letters (^a^, ^b^) indicate significant differences compared with the other groups, as assessed using one-way ANOVA (normal distribution) and the Kruskal–Wallis test (non-normal distribution) (*p* < 0.05). ^2^ T2DM: type 2 diabetes mellitus; BMI: body mass index; SBP: systolic blood pressure; DBP: diastolic blood pressure; BF%: body fat percentage; SMI: skeletal muscle mass index; ASM: appendicular skeletal muscle; HGS: handgrip strength; SPPB: short physical performance battery; CC: calf circumference; SARC-F: questionnaire for screening sarcopenia; PA: physical activity was assessed by the Godin leisure-time exercise questionnaire.

**Table 6 nutrients-15-03892-t006:** Multivariate logistic regression analysis of select risk factors associated with possible sarcopenia or sarcopenia ^1,2^.

Parameters	Nagelkerke R-Square	Odds Ratio (95% CI)	*p*
**Possible sarcopenia**	0.210		
Serum 25(OH)D (ng/mL)		0.927 (0.863–0.997)	0.040
Energy (per kcal/kg BW)		0.877 (0.774–0.994)	0.040
**Sarcopenia**	0.890		
Age (per year)		1.231 (1.071–1.415)	0.004
BMI (kg/m^2^)		0.438 (0.283–0.680)	0.000
Serum 25(OH)D (ng/mL)		0.901 (0.812–0.999)	0.048
Protein intake (per g/kg BW)		0.001 (0.000–0.166)	0.009
Vitamin D intake (per μg)		0.915 (0.849–0.986)	0.020

^1^ Significant difference by multivariate logistic regression with the stepwise method (*p* < 0.05). ^2^ CI: confidence interval; 25(OH)D: 25-hydroxyvitamin D; BMI: body mass index; BW: body weight.

## Data Availability

The data presented in this study are available on request from the corresponding author. The data are not publicly available due to ethical restrictions.

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
