# Peer review of "Association of Possible Sarcopenia or Sarcopenia with Body Composition, Nutritional Intakes, Serum Vitamin D Levels, and Physical Activity among Patients with Type 2 Diabetes Mellitus in Taiwan"

_nutrients, 2023, doi:10.3390/nu15183892_

Round 1

Reviewer 1 Report (New Reviewer)

Major comments

1.     The authors indicate that the prevalence of sarcopenia among patients with T2DM is 37% (lines 296-300). I do not strongly agree with this statement. First of all, the sample is too small to estimate a prevalence (this sample size would lead to an estimate with an error bound of 10%). Also, the authors have not indicated whether this sample is a sample of diabetics from the entire population of Taiwan. In any case, I very much doubt that this is the case. In addition, a portion of the possible sarcopenias will surely be sarcopenias, which would further raise the prevalence estimated by the authors. It could be as high as 50% and this is not at all reliable.

2.     The statistical analysis es poor. The following sections highlight some of the shortcomings.

a.     The authors should distinguish between “parameters” and “variables”. For example, the mean of a variable is a parameter (fixed quantity) that summarizes it. Therefore, in tables 1 and 2, “parameter” should be replaced by “variable”.

b.    The “duration of diabetes” is a variable that evidently deviates strongly from the normality hypothesis. In these cases, the variable should be summarized as median and interquartile range. When the assumptions of normality are satisfied, 95% of the observations are between the mean plus/minus twice the standard deviation, which is obviously not the case. In tables 4 and 5, at least the “Vitamin-D” variable does not satisfy the hypotheses of normality in each of the groups. The authors can explore the normality of the variables by means of QQ-plots.

c.     The chi-square test is an asymptotic test and the p-value is spurious for small values in the cells. In such cases, Fisher's exact test should be used. This is the case of the “Insulin injection” variable. The authors used the chi-square test and obtained a p-value of 0.077. The p-value corresponding to Fisher's exact test is 0.064. This should be corrected for other variables such as “Betel nut”.

d.    The logistic models given in Tables 6 and 7 are erratic. In general, regression models should be constructed by selecting variables that could potentially be associated with the  outcome. In the model in Table 6, of the four covariates included, only one maintains a significant association with outcome, which means that the model is over-fitted and therefore spurious. The authors should introduce into the logistic analysis the potential risk factors for sarcopenia (or possible sarcopenia) and then carry out a variable selection. I recommend the method of the “best subset regression” in conjunction with the Akaike information criterion (AIC).

Minor comments

1.     Line 16 (abstract). The objective indicated in these lines does not coincide with the study developed. The correct one would be: “This study aims to evaluate the association between sarcopenia and blood biochemical parameters …. in patients with type 2 diabetes mellitus (T2DM)”. 

2.     Line 26: Change “BMI” by “body mass index (BMI)”

3.     Lines 48-50. All subjects included in this study are diabetic. Therefore, in this study it is not possible to identify factors associated with T2DM.

4.     The style of the text is poor. For example, in line 61, between "Taiwan" and "participants" there should be a period and not a comma.

5.     Line 70. The expression “or losing >5% of body weight” is inadequate. It should be: “or a loss of more than 5% of body weight”.

6.     Line 83. The term "experiment" is inappropriate. This is not an experimental study.

7.     Line 162. Delete the term “coefficients”.

8.     Line 163. A Cronbach's alpha of 0.715 is only acceptable, not good.

9.  Tables 6 y 7. Change “Nagelkerkr” by “Nagelkerke”. This statistic should be referred to in the "Statistical Analysis" subsection.

The style of the text is poor. For example, in line 61, between "Taiwan" and "participants" there should be a period and not a comma.

Author Response

1. The authors indicate that the prevalence of sarcopenia among patients with T2DM is 37% (lines 296-300). I do not strongly agree with this statement. First of all, the sample is too small to estimate a prevalence (this sample size would lead to an estimate with an error bound of 10%). Also, the authors have not indicated whether this sample is a sample of diabetics from the entire population of Taiwan. In any case, I very much doubt that this is the case. In addition, a portion of the possible sarcopenias will surely be sarcopenias, which would further raise the prevalence estimated by the authors. It could be as high as 50% and this is not at all reliable.

Response:

Thank you for your precious comments and advice, we remove some sentences regarding the prevalence of sarcopenia and re-wrote the sentence in the revised manuscript.

2. The statistical analysis es poor. The following sections highlight some of the shortcomings.

a. The authors should distinguish between “parameters” and “variables”. For example, the mean of a variable is a parameter (fixed quantity) that summarizes it. Therefore, in tables 1 and 2, “parameter” should be replaced by “variable”.

b. The “duration of diabetes” is a variable that evidently deviates strongly from the normality hypothesis. In these cases, the variable should be summarized as median and interquartile range. When the assumptions of normality are satisfied, 95% of the observations are between the mean plus/minus twice the standard deviation, which is obviously not the case. In tables 4 and 5, at least the “Vitamin-D” variable does not satisfy the hypotheses of normality in each of the groups. The authors can explore the normality of the variables by means of QQ-plots.

Response:

Thank you for your precious comments and advice. More detailed statistical analysis was added on page 4-5.

We have replaced ‘parameters’ with ‘variables’; we also checked the normality of the variables and modified the data in the revised manuscript.

c.The chi-square test is an asymptotic test and the p-value is spurious for small values in the cells. In such cases, Fisher's exact test should be used. This is the case of the “Insulin injection” variable. The authors used the chi-square test and obtained a p-value of 0.077. The p-value corresponding to Fisher's exact test is 0.064. This should be corrected for other variables such as “Betel nut”.

Response:

Thank you for your suggestion, we re-wrote some p-values by using Fisher’s exact test when the frequency of events in a certain cell was low.

d. The logistic models given in Tables 6 and 7 are erratic. In general, regression models should be constructed by selecting variables that could potentially be associated with the outcome. In the model in Table 6, of the four covariates included, only one maintains a significant association with outcome, which means that the model is over-fitted and therefore spurious. The authors should introduce into the logistic analysis the potential risk factors for sarcopenia (or possible sarcopenia) and then carry out a variable selection. I recommend the method of the “best subset regression” in conjunction with the Akaike information criterion (AIC).

Response:

Thank you for your precious comments and advice, we have revised the manuscript.

Due to the sample size needed for stable estimates, we eventually combined the possible sarcopenia group into a single sarcopenia group, that is, the binary logistic regression was used to predict the potential risk factors of sarcopenia.

Minor comments

  1. Line 16 (abstract). The objective indicated in these lines does not coincide with the study developed. The correct one would be: “This study aims to evaluate the association between sarcopenia and blood biochemical parameters …. in patients with type 2 diabetes mellitus (T2DM)”. 
  2. Line 26: Change “BMI” by “body mass index (BMI)”
  3. Lines 48-50. All subjects included in this study are diabetic. Therefore, in this study it is not possible to identify factors associated with T2DM.
  4. The style of the text is poor. For example, in line 61, between "Taiwan" and "participants" there should be a period and not a comma.
  5. Line 70. The expression “or losing >5% of body weight” is inadequate. It should be: “or a loss of more than 5% of body weight”.
  6. Line 83. The term "experiment" is inappropriate. This is not an experimental study.
  7. Line 162. Delete the term “coefficients”.
  8. Line 163. A Cronbach's alpha of 0.715 is only acceptable, not good.
  9.  Tables 6 y 7. Change “Nagelkerkr” by “Nagelkerke”. This statistic should be referred to in the "Statistical Analysis" subsection.

Response:

Thank you for all your suggestions, we have revised the manuscript.

Reviewer 2 Report (New Reviewer)

This an interesting manuscript. However, some points should be addressed.

Patient recruitment details were not specified in the abstract. add, please.

The abstract lacks measures of association such as odds ratios and 95% confidence intervals. 

Regarding BMI measurement using bioelectrical impedance analysis (BIA) by Inbody 270 (Biospace Co. Ltd., Seoul, Korea), the sentence needs revision to clarify how BMI was obtained through BIA.

In line 111, the authors should explicitly state the definition used for sarcopenia in the study and provide a detailed explanation of how participants were categorized into the non-sarcopenia, possible sarcopenia, and sarcopenia groups.

The sentence at lines 177-180 is inaccurate as it suggests using one-way analysis of variance (ANOVA) for all variables, including categorical ones. This should be corrected, and the appropriate statistical methods for each type of variable should be specified. 

Additionally, in Table 1, the authors should include information about the measured covariates (smoking, alcohol, etc.) in the methods section.

Furthermore, in Tables 6 and 7, the number of participants included in each regression analysis should be clearly indicated in the table heading.

Lastly, there is an issue with the interpretation of odds ratios for dietary protein intake and dietary energy intake. The odds ratio (OR) is mistakenly presented as "<0.001," which is not a valid numerical value. This should be corrected to provide the actual numerical value of the odds ratio along with the 95% confidence interval and p-value.

Author Response

  1. This an interesting manuscript. However, some points should be addressed. Patient recruitment details were not specified in the abstract. add, please.
  1. The abstract lacks measures of association such as odds ratios and 95% confidence intervals.

Response:

We apologize for not being able to describe the information on patient recruitment and ORs (95% CI) very clearly due to the word limit. We re-wrote the sentence in the revised manuscript.

  1. Regarding BMI measurement using bioelectrical impedance analysis (BIA) by Inbody 270 (Biospace Co. Ltd., Seoul, Korea), the sentence needs revision to clarify how BMI was obtained through BIA.

Response:

As we mentioned in the manuscript, height (cm) was collected using a stadiometer. BMI is calculated and displayed on the screen after the patient’s height is entered and body weight (kg) is measured by the InBody 270. Therefore, we didn’t make any changes to the manuscript.

  1. In line 111, the authors should explicitly state the definition used for sarcopenia in the study and provide a detailed explanation of how participants were categorized into the non-sarcopenia, possible sarcopenia, and sarcopenia groups.

Response:

Thank you for your suggestion, we have added a more detailed interpretation regarding sarcopenia diagnostic criteria on page 3 (lines 139-146). According to the criteria of AWGS 2019 (Chen L.K., 2020 [5], as shown in the file), possible sarcopenia was defined as low muscle strength or low physical performance. Sarcopenia was defined as low skeletal muscle mass and low muscle strength or/and low physical performance.

  1. The sentence at lines 177-180 is inaccurate as it suggests using one-way analysis of variance (ANOVA) for all variables, including categorical ones. This should be corrected, and the appropriate statistical methods for each type of variable should be specified.

Response:

Thanks for your kind reminder, more detailed statistical analysis was added on page 4-5.

We have revised the manuscript in the statistical section regarding the statistical methods for continuous variables, categorical variables, normal distribution, and non-normal distribution.

  1. Additionally, in Table 1, the authors should include information about the measured covariates (smoking, alcohol, etc.) in the methods section.

Response:

As we mentioned in the manuscript (lines 75-81), we asked participants to complete the general questionnaire, which included questions related to sex, age, diabetes onset period, lifestyle habits (smoking, alcohol drinking, and betel nut consumption), medications used in the treatment of chronic diseases (such as oral hypoglycemic agents, insulin injection, blood-pressure-lowering agents, and hypolipidemic agents).

  1. Furthermore, in Tables 6 and 7, the number of participants included in each regression analysis should be clearly indicated in the table heading.

Response:

Thank you for your suggestion, we have added the number of participants in the table heading.

  1. Lastly, there is an issue with the interpretation of odds ratios for dietary protein intake and dietary energy intake. The odds ratio (OR) is mistakenly presented as "<0.001," which is not a valid numerical value. This should be corrected to provide the actual numerical value of the odds ratio along with the 95% confidence interval and p-value.

Response:

Thank you for your kind reminder, we have revised the manuscript. Due to the sample size needed for stable estimates, we eventually combined the possible sarcopenia group into a single sarcopenia group, that is, the binary logistic regression was used to predict the potential risk factors of sarcopenia.

Reviewer 3 Report (New Reviewer)

This manuscript is an observational study of correlates of biochemical and anthropomorphic parameters associated with sarcopenia among older Taiwanese with type 2 diabetes.

Recommendations for clarification in the methodology, reporting of results, and the conclusions are provided to the authors.

This manuscript is an observational study of correlates of biochemical and anthropomorphic parameters associated with sarcopenia among older Taiwanese with type 2 diabetes.

Methods – enrollment -please give information on which clinics, estimate total population for which sample was drawn, and provide this information in section 2.1 and also add to Figure 1 enrollment table.

Methods–please clarify the cut off on the SARC–F score for diagnosis of sarcopenia, section 2.4

Methods–please indicate whether vitamin D2 or D3 level was measured.  Standard measurement for vitamin D status is 25-hydroxy vitamin D (D2).  If vitamin D3 is measured please indicate why.  This may affect information on lines 24, 42, 138, 146, 320, and 368 as well as tablets 3, 4 and 6.

Conclusion–all correlations are associations.  The findings are overstated.  Restate that the major findings may have potential for population health in identifying and preventing sarcopenia.

Author Response

  1. This manuscript is an observational study of correlates of biochemical and anthropomorphic parameters associated with sarcopenia among older Taiwanese with type 2 diabetes. Recommendations for clarification in the methodology, reporting of results, and the conclusions are provided to the authors.
  2. This manuscript is an observational study of correlates of biochemical and anthropomorphic parameters associated with sarcopenia among older Taiwanese with type 2 diabetes.

Methods – enrollment -please give information on which clinics, estimate total population for which sample was drawn, and provide this information in section 2.1 and also add to Figure 1 enrollment table.

Response:

Thank you for your suggestion. However, our recruitment process was: we recruited participants in a primary care clinic after performing a G*Power analysis, and this study had to include 110 T2DM patients. Both the inclusion and exclusion criteria were applied at the same time to screen and determine the eligibility of individuals for participation in this study. After the enrollment of 110 participants, the recruitment was terminated.

In addition, we also found it difficult to recruit participants due to our strict inclusion criteria, and the the number of 110 also the number of participants that our primary care clinic can recruit. Therefore, we didn’t make any changes to the manuscript.

  1. Methods–please clarify the cut off on the SARC–F score for diagnosis of sarcopenia, section 2.4

Response:

According to the criteria of AWGS 2019 (Chen L.K., 2020 [5], as shown in the picture, it was also a reply to reviewer 2 in the file), the cut-off on the SARC-F score was ≥ 4 (section 2.3), so we didn’t make any changes.

  1. Methods–please indicate whether vitamin D2 or D3 level was measured. Standard measurement for vitamin D status is 25-hydroxy vitamin D (D2). If vitamin D3 is measured please indicate why.  This may affect information on lines 24, 42, 138, 146, 320, and 368 as well as tablets 3, 4 and 6.

Response:

Thanks for your kind reminder, we measured the level of 25(OH)D for vitamin D status, and we have revised the manuscript.

  1. Conclusion–all correlations are associations. The findings are overstated. Restate that the major findings may have potential for population health in identifying and preventing sarcopenia.

Response:

Thank you for your suggestion, we re-wrote the sentence in the revised manuscript.

Round 2

Reviewer 1 Report (New Reviewer)

Major comments

In order to identify factors with an independent association with sarcopenia, the authors combine the categories “sarcopenia/possible sarcopenia” into a single group. I have serious doubts about this categorization. First, of the 110 patients included in the study, 72 (65.5%) would be assimilated to sarcopenia, which does not seem reliable. Secondly, in terms of anthropometric measurements, the group of possible sarcopenias more closely resembles that of non-sarcopenia. I think it is more correct to keep the comparisons of the first version, namely: possible sarcopenia vs. non-sarcopenia (Logistic model 1) and sarcopenia vs. non-sarcopenia (Logistic model 2).

Minor comments

1.     Ln 53-54. The authors aim to (sic) “This study aimed (i) to understand the prevalence of possible sarcopenia and sarcopenia”. What is the reference population in which these prevalences are estimated?

2.     Ln. 60. “partic-ipants“. I think that the syllables at the end of a line cannot be separated.

3.     Ln 67. Change “stand/ walk” to “stand or walk”.

4.     Ln. 82. Change “process, and precautions” to “process and precautions” (delete the comma).

5.     Ln 83-84. Change: “The study was conducted according to the principles presented in the Declaration of Helsinki” to “The study was conducted according to the Declaration of Helsinki”.

6.     Ln 182-184. I think this paragraph could be improved as follows: “Categorical variables are expressed as frequencies and percentages and continuous as mean and standard deviation (SD) when data followed a normal distribution, or as median and interquartile range (IQR = 25th – 75th percentile) when distribution departed from normality”. In each group, normality was tested using the Kolmogorov-Smirnov test.

7.     Ln 187-188. Change: “Categorical variables were compared using the Chi-square test or Fisher’s exact test” to “The percentages were compared, as appropriate, using the Chi-square test or the exact Fisher test”.

8.     Ln 189-190. The following paragraph is wrong: “Due to the sample size needed for stable estimates, the Possible sarcopenia group was merged into a single Sarcopenia group”. The following paragraph is more correct: “Given the low sample size, the sarcopenia and possible sarcopenia groups were merged into a single category”. Then, a logistic model for the outcome sarcopenia/possible sarcopenia versus no sarcopenia” was estimated.

9.     Ln 205. Change “46 male and 64 female” to “46 males and 64 females”.

10.  Table 2. The number of patients with sarcopenia without daily sun exposure is 7, not 8.

11.  The association between sarcopenia groups and daily sun exposure times should be tested with Fisher's exact test, not with the chi-square test.

12.  Ln 232 (Table footer). Change: “number of people (%)” to “frequencies (%)”. Make this change in all tables.

13.  Table 3. Change “parameters” to “variables”.

14.  Table 5. Change: “Muscle Mass, Strength, and Performance” to “Muscle Mass, Strength, and Performance” (delete the comma).

15.  Table 6. Change “(year)” to “(per year)”, “(Kg)” to “(per Kg)” and “(g/Kg BW)” to “(per g/Kg BW)”.

16.  Ln 368-370. The authors state that (sic): “The study found that adequate dietary protein consumption and higher body weight and serum vitamin D levels might prevent sarcopenia”. This statement should be qualified. The study is not a clinical trial and therefore, it is not appropriate to establish an effect of protein intake as a protective factor for sarcopenia. In any case, there is a clear statistical association. The authors should investigate this point further.

 In general, the style of the manuscript is poor and should not be published in its present state. The authors should ask an expert to revise it.

Author Response

Responses to the comments of Reviewer #1

  1. In order to identify factors with an independent association with sarcopenia, the authors combine the categories “sarcopenia/possible sarcopenia” into a single group. I have serious doubts about this categorization. First, of the 110 patients included in the study, 72 (65.5%) would be assimilated to sarcopenia, which does not seem reliable. Secondly, in terms of anthropometric measurements, the group of possible sarcopenias more closely resembles that of non-sarcopenia. I think it is more correct to keep the comparisons of the first version, namely: possible sarcopenia vs. non-sarcopenia(Logistic model 1) and sarcopenia vs. non-sarcopenia (Logistic model 2).

Response:

Thank you for your precious comments and advice, we have modified Table 6 according to the comment and revised the manuscript.

Minor comments

  1. Ln 53-54. The authors aim to (sic) “This study aimed (i) to understand the prevalence of possible sarcopenia and sarcopenia”. What is the reference population in which these prevalences are estimated?
  2. 60. “partic-ipants“. I think that the syllables at the end of a line cannot be separated.
  3. Ln 67. Change “stand/ walk” to “stand or walk”.
  4. 82. Change “process, and precautions” to “process and precautions” (delete the comma).
  5. Ln 83-84. Change: “The study was conducted according to the principles presented in the Declaration of Helsinki” to “The study was conducted according to the Declaration of Helsinki”.
  6. Ln 182-184. I think this paragraph could be improved as follows: “Categorical variables are expressed as frequencies and percentages and continuous as mean and standard deviation (SD) when data followed a normal distribution, or as median and interquartile range (IQR = 25th – 75th percentile) when distribution departed from normality”. In each group, normality was tested using the Kolmogorov-Smirnov test.
  7. Ln 187-188. Change: “Categorical variables were compared using the Chi-square test or Fisher’s exact test” to “The percentages were compared, as appropriate, using the Chi-square test or the exact Fisher test”.
  8. Ln 189-190. The following paragraph is wrong: “Due to the sample size needed for stable estimates, the Possible sarcopenia group was merged into a single Sarcopenia group”. The following paragraph is more correct: “Given the low sample size, the sarcopenia and possible sarcopenia groups were merged into a single category”. Then, a logistic model for the outcome sarcopenia/possible sarcopenia versus no sarcopenia” was estimated.
  9. Ln 205. Change “46 male and 64 female” to “46 males and 64 females”.
  10. Table 2. The number of patients with sarcopenia without daily sun exposure is 7, not 8.
  11. The association between sarcopenia groups and daily sun exposure times should be tested with Fisher's exact test, not with the chi-square test.
  12. Ln 232 (Table footer). Change: “number of people (%)” to “frequencies (%)”. Make this change in all tables.
  13. Table 3. Change “parameters” to “variables”.
  14. Table 5. Change: “Muscle Mass, Strength, and Performance” to “Muscle Mass, Strength, and Performance” (delete the comma).
  15. Table 6. Change “(year)” to “(per year)”, “(Kg)” to “(per Kg)” and “(g/Kg BW)” to “(per g/Kg BW)”.
  16. Ln 368-370. The authors state that (sic): “The study found that adequate dietary protein consumption and higher body weight and serum vitamin D levels might prevent sarcopenia”. This statement should be qualified. The study is not a clinical trial and therefore, it is not appropriate to establish an effect of protein intake as a protective factor for sarcopenia. In any case, there is a clear statistical association. The authors should investigate this point further.

Response:

Thank you for all your suggestions, we have revised the manuscript.

Comments on the Quality of English Language

 In general, the style of the manuscript is poor and should not be published in its present state. The authors should ask an expert to revise it.

Response:

We apologize for the language problems in the original manuscript. The initial version has been edited by “Wallace Academic Editing”, and we are currently continuing to engage their services for further editing. Due to the editing timeline, it may take a few more days. Once we receive the revised version, we will send it to you and the Nutrients Editorial Office.

Reviewer 2 Report (New Reviewer)

My comments have been addressed by the authors. I have no further comments to add. 

Author Response

Responses to the comments of Reviewer #2

My comments have been addressed by the authors. I have no further comments to add. 

Response:

Thanks for your approval.

Reviewer 3 Report (New Reviewer)

Version 2 of the manuscript has greatly enhanced the clarity of the presentation.

Suggestions for revision provided to the authors in the attachment.

Author Response

Responses to the comments of Reviewer #3

Version 2 of the manuscript has greatly enhanced the clarity of the presentation.

Suggestions for revision provided to the authors in the attachment.

Response:

Thank you for all your suggestions and modifications, we have revised the manuscript.

This manuscript is a resubmission of an earlier submission. The following is a list of the peer review reports and author responses from that submission.

Round 1

Reviewer 1 Report

The title is clear. 

Abstract: Line 22-24 - Comparisons may not be highlighted in the abstract. 

The introduction is clear. The second objective may not be needed and not reflected in the title. 

Section 3.8 Statistical analysis - Multiple logistic regression methodology needs to be expanded, to include assumptions etc.

Results section requires major rewriting. As the focus of the paper is to highlight the prevalence and the risk factors, the results section should focus on these 2 aspects. Present binary log reg with crude ORs, show the selection of variables for multivariate analysis before presenting multiple log reg with adjusted ORs. 

Discussion should include a discussion on strengths and limitations of this study. 

Author Response

  1. Abstract: Line 22-24 - Comparisons may not be highlighted in the abstract.

Response: Thank you for your suggestion, we deleted those sentences.

  1. The introduction is clear. The second objective may not be needed and not reflected in the title.

Response: Thank you for your comments, we apologize for not expressing ourselves clearly. The precedent version of the title has been changed, so we kept the second objective.

  1. Section 3.8 Statistical analysis - Multiple logistic regression methodology needs to be expanded, to include assumptions etc.

Response: Thank you for your suggestion, we modified our manuscript.

  1. Results section requires major rewriting. As the focus of the paper is to highlight the prevalence and the risk factors, the results section should focus on these 2 aspects. Present binary log reg with crude ORs, show the selection of variables for multivariate analysis before presenting multiple log reg with adjusted ORs.

Response: Thank you for your comments, we apologize for not expressing ourselves clearly. The precedent version of the title has been changed. We also have revised the corresponding part of the manuscript on the methods section (Statistical analysis) to address the multivariable logistic regression we used.

  1. Discussion should include a discussion on strengths and limitations of this study.

Response: Thank you for your suggestion, we added this information to page 10.

Reviewer 2 Report

Dear Authors,

The research is well designed on an important topic,however the results are already known and well studied in the literature, that, I could not find anything new.

Best regards

Author Response

  1. The research is well designed on an important topic, however the results are already known and well studied in the literature, that, I could not find anything new.

Response: In our revisions, we have attempted to modify the title and we hope those changes have been made to make the expression more precise and more accurate. We also added some strengths of this study to our manuscript on page 10.

Reviewer 3 Report

This is an interesting  article where the authors concluded that “Our study indicated that higher BMI, dietary protein intake and serum 27 vitamin D3 levels, and lower percent of body fat might prevent the occurrence of possible sarcopenia 28 and sarcopenia in elderly patients with T2DM

However in order to be published in this prestigious journal, authors must be included the next changes:

Material and method

Please, it is necessary to add  the device to determine biochemical parameters (Brad, City and State). These data of scale and stadiometer must be included (too).

Main variable used to calculate sample size must be included.

SGlt2 drus are excluded in this study.. See the recent article (Effect of SGLT-2 inhibitors on body composition in patients with type 2 diabetes mellitus: A meta-analysis of randomized controlled trials Runzhou PanID, Yan Zhang, Rongrong Wang, Yao Xu, Hong Ji, Yongcai ZhaoID* Department of Endocrinology, Cangzhou Central Hospital, Cangzhou, Hebei Province, Chin)

Discussion            

Please, authors need to discuss the role of different drugs of DM2 in its results, and the role of insulin therapy as an anabolic treatment.

Limitation section must be included.

A section of strengths should be included

Author Response

  1. Please, it is necessary to add the device to determine biochemical parameters (Brad, City, and State). These data of scale and stadiometer must be included (too).

Response: Thank you for your suggestion, we have revised the corresponding part of the manuscript in the methods section.

  1. Main variable used to calculate sample size must be included.

Response: Thank you for your suggestion, we modified our manuscript.

  1. SGlt2 drus are excluded in this study.. See the recent article (Effect of SGLT-2 inhibitors on body composition in patients with type 2 diabetes mellitus: A meta-analysis of randomized controlled trials Runzhou PanID, Yan Zhang, Rongrong Wang, Yao Xu, Hong Ji, Yongcai ZhaoID* Department of Endocrinology, Cangzhou Central Hospital, Cangzhou, Hebei Province, Chin)

Please, authors need to discuss the role of different drugs of DM2 in its results, and the role of insulin therapy as an anabolic treatment.

Response: Thank you for your comments, your suggestion is valid. However, in this cross-sectional study, we only recorded the prescriptions and categorized them with oral medications and insulin injections in our healthcare system. Metformin is the first-line oral medication, so we may have misunderstood the meaning when we wrote the manuscript. It is our negligence and we are very sorry about this, we corrected it to “Oral medications”. The detail of prescriptions and the correlation between drugs/insulin therapy and sarcopenia are also one of the research limitations in our article.

Regarding SGLT-2 inhibitors, according to Taiwan’s National Health Insurance system, the payment standard requires HbA1c>8% to be tested twice within six months and most of our participants in this study did not meet the standard, so they are not included in the discussion. Meanwhile, we feel that the scope of work of the present paper can support our conclusions and no significant differences between the three groups were discovered in those medications and injections. Therefore, we suggest that the additional discussion could be included in a follow-up paper.

  1. Limitation section must be included. A section of strengths should be included

Response: Thank you for your suggestion, we added this information to page 10.

Round 2

Reviewer 1 Report

While the other sections have been updated according to suggestion, the logistic regression part still lacking the details I have requested earlier. Associations are best shown in regression analysis. Hence, it is important this is shown clearly. 

Present binary log reg with crude ORs, show the selection of variables for multivariate analysis before presenting multiple log reg with adjusted ORs.

You also need to show the selection of variables from binary log reg to multiple log reg - how the variables are chosen? what are those variables? What were the assumptions made in multivariate analysis? Were there any checking on interactions? multicollinearity? Hosmer test? what is the R2 value for the multivariate model? 

Author Response

Responses to the comments of Reviewer #1

1. While the other sections have been updated according to suggestion, the logistic regression part still lacking the details I have requested earlier. Associations are best shown in regression analysis. Hence, it is important this is shown clearly.

Present binary log reg with crude ORs, show the selection of variables for multivariate analysis before presenting multiple log reg with adjusted ORs.

You also need to show the selection of variables from binary log reg to multiple log reg - how the variables are chosen? what are those variables? What were the assumptions made in multivariate analysis? Were there any checking on interactions? multicollinearity? Hosmer test? what is the R2 value for the multivariate model?

Response:

Thank you for your suggestion, we apologize for missing some important information in our Statistical analysis, which we have revised in the manuscript. We also modified the data (including multicollinearity and Hosmer-Lemeshow test) in Table 6 & 7 on page 8-9.

We used the univariate logistic regression analysis first and to explore the factors associated with the presence of possible sarcopenia (4 factors associated with possible sarcopenia: Age, serum VD level, protein intake, and percentage of body fat) and sarcopenia (6 factors associated with sarcopenia: Age, serum LDL-C level, energy & protein intake, BMI and percentage of body fat). We then used multivariate logistic regression to reveal the final factors, as showed in Table 6 & 7.

Reviewer 2 Report

Thank you for the revisions.

Minör editing is required.

Author Response

Responses to the comments of Reviewer #2

  1. Comments on the Quality of English Language Minör editing is required.

 Response:

Thank you for your suggestion. We revised some text in the manuscript and upload our “English Editing Certification” by Wallace Academic Editing.

Reviewer 3 Report

Dear Authors

After reviewing the comments of the reviewers and reading the manuscript and the corrections made by the authors. We accept this manuscript to be published.

We are happy for the decision taken and we appreciate the effort made by the authors

Best regards,

Daniel de Luis

Author Response

Responses to the comments of Reviewer #3

1.After reviewing the comments of the reviewers and reading the manuscript and the corrections made by the authors. We accept this manuscript to be published.

We are happy for the decision taken and we appreciate the effort made by the authors

Response:

We appreciate for your review and suggestion, thank you.